# Compulsory School Achievement and Future Gambling Expenditure: A Finnish Population-Based Study

**DOI:** 10.3390/ijerph19159444

**Published:** 2022-08-01

**Authors:** Tiina Latvala, Anne H. Salonen, Tomi Roukka

**Affiliations:** 1Finnish Institute for Health and Welfare, 00250 Helsinki, Finland; anne.salonen@thl.fi (A.H.S.); tomi.roukka@utu.fi (T.R.); 2Economics Department, Turku School of Economics, University of Turku, 20500 Turku, Finland

**Keywords:** gambling, gambling expenditure, education, population study, register data, school achievement, grade point average (GPA), socio-economic position

## Abstract

Background: Gambling is associated with many conditions that can compromise young people’s health and wellbeing, such as substance use and poor school achievement. Conversely, low school achievement can be linked to lower socio-economic position. Thus, the aim of this study is to examine whether compulsory school achievement is linked with gambling participation and gambling expenditure (GE) later in youth and whether GE is linked with lower socio-economic position. Methods: The Finnish Gambling Harms survey data (*n* = 7186) were used. The data were collected in three regions during spring 2017. Participants aged 18–29 years old were selected from the data. Past-year GE was examined using two measures: weekly gambling expenditure (WGE, in €) and relative gambling expenditure (RGE, in %). Logistic regression and log-linear regression models for past-year gambling, WGE and RGE were created. Results: Persons who had no more than a mediocre grade point average (GPA) had a 25% higher WGE and 30% higher RGE in 2016 than those who had an outstanding GPA in the compulsory school. Compared with persons with an outstanding GPA, those with a satisfactory to very good GPA spent 13% more on gambling, and their RGE was 17% higher. Additionally, those with lower socio-economic status (SES) had a higher WGE and RGE compared with higher SES. Conclusions: Even after controlling for other crucial background characteristics, early life success, in the form of compulsory school outcomes, seems to correlate with gambling expenditures later in youth. This suggests that the gambling behaviour can be linked to the cognitive ability of an individual. Our findings also imply that gambling could be more heavily concentrated on individuals that are already more socially disadvantaged. However, it is worth noting that individual factors such as traumas, antisocial personality, anxiety and depression are all associated with gambling and poor academic achievement. Overall, this suggests that various educational tools at a younger age can be effective in preventing gambling-related problems in later life.

## 1. Introduction

Gambling is a significant public health concern [1,2]. Gambling is associated with many conditions that can compromise young people’s health and wellbeing, such as depression, substance use, delinquency and poor school achievement [3,4,5,6,7]. Risky behaviours, such as substance use and gambling, are often higher among younger adults than among their older counterparts [8,9,10]. Problem gambling prevalence rates vary between 0.1 and 5.8% of adults worldwide and between 0.1% and 3.4% in Europe [11]. Among 18–34-year-old Finns, the problem gambling prevalence rate (SOGS 3+) varied from 4.8% to 5.3% in 2019, while the corresponding figure for the whole population was 3.0% [12].

A high gambling frequency and participation in multiple game types are typically associated with high gambling expenditure (GE) [13,14,15]. Internationally, high GE is very concentrated on a small group of gamblers [12,13,16]. In fact, depending on time and survey methods, 2% to 5% of gamblers produce half (50%) of the total GE in Finland [12]. A significant proportion of the GE is produced by persons with at-risk and problem gambling patterns [13]. Despite this, high GE does not automatically indicate harmful gambling behaviour; however, the link exists [17,18,19].

Internationally, men tend to spend more money on gambling than women [20,21,22], in addition to which gambling problems tend to be more prevalent among men [23,24,25]. Furthermore, younger people have been identified as a risk group for developing a gambling problem [23,26]. Despite that, gambling expenditure is lower among younger adults compared with their older counterparts [15,20]. Moreover, GE varies within genders and age groups. In 2015, while one in four (26.9%) of men’s GE came from 25 to 34-year-olds, the corresponding figure for women was 7.5% [20]. Overall, high GE among females is more centred for older age groups in Finland [13,20] Internationally, both younger and older male adults spend more money on gambling than women [20,22].

Several researchers have identified social disparity and health inequalities related to gambling [27,28,29]. Persons with lower income spend more in relation to their net income than those with higher income [20]. Gamblers in the lowest GE group differed from those in the intermediate and highest GE group in terms of their socio-demographic background and gambling behaviour [13]. Additionally, low school achievement and a low level of education have been linked with at-risk and problem gambling [3,4,5,26]. Conversely, low school achievement can be linked to lower socio-economic position, such as unemployment, lower education and incomes [30]. On the other hand, socio-economic position is associated with an individual’s school achievement, as individuals with higher socio-economic position, on average, perform better than children from lower positions [31].

There is broad agreement that there is a moderate to strong correlation between cognitive ability and school achievement [32]. The cognitive ability of an individual is also linked to gambling participation as well as problem and pathological gambling [33,34]. The higher the premorbid cognitive ability or “intelligence”, the lower the probability that a person engages in gambling. However, the role of social factors appears to be larger for gambling propensity than for a cognitive factor. Chen, Chiu, Smith and Yamada [35] discuss how GPA, as a measurement of cognitive ability, also reflects the intrinsic motivation of the individual. Although GPA is not in any sense a perfect measure of cognitive ability, previous studies have found it being related with consistency in the social preferences as well as in risk preferences [35,36]. This, on the other hand, suggests that a connection should emerge between an individual’s GPA and gambling decisions.

Studies have highlighted the need for global prevention efforts that reduce risk factors for problem gambling and screen young people with high-risk profiles [26]. However, to our knowledge, there is a limited amount of literature on gambling, school achievement and social disparity among youth. Nevertheless, three hypotheses are tested:(1)Compulsory school achievement, measured as the GPA of theoretical subjects, correlates with gambling participation and gambling expenditure later in life;(2)Higher gambling expenditure is linked with lower socio-economic position among youth;(3)Higher gambling expenditure is linked with the male gender among youth.

## 2. Materials and Methods

### 2.1. Participants

The Finnish Institute for Health and Welfare was responsible for conducting the population-based Gambling Harms Survey [37]. The data were collected by Statistics Finland between January and March in 2017 among residents from three geographical areas: Uusimaa, Pirkanmaa and Kymenlaakso. This area covers 42% of the Finnish population. The data were collected by online and postal surveys available in two official languages: Finnish and Swedish. A total of 20,000 potential participants, 18 years old or over, who understand Finnish or Swedish, were randomly selected from the population register.

An invitation letter was sent to the potential participant’s home address, in which they received written information about the study and the principles of voluntary participation [38]. Participants were informed that the study involved the register linkage and their statutory right to disclose data for scientific purposes. Information about the register and a list of the register-based variables used in the analyses were also described in the letter. Two reminder letters were sent, including postal questionnaires and prepaid return envelopes. Further, an invitation letter included a link to the online survey with a personal participation code.

Institutionalised persons (prisoners, infirmed, etc.) and non-eligible (*n* = 67) respondents were excluded, leaving 19,933 potential participants. Overall, 7186 adults participated in the study, which gave a response rate of 36.1%. Of the participants, 71% (*n* = 5084) participated in the online survey and 29% (*n* = 2102) in the postal survey [38]. Almost half (47.7%) were males, and the ages ranged from 18 to 94 years (M = 50.5, SD = 18.8). For the purposes of this study, only 18–29-year-old respondents were selected (*n* = 1334). Younger respondents, especially 18–24-year-old men, were the least active in participating in the study. This was also the case for divorced or single respondents and respondents with lower education [38].

### 2.2. Measures from the Survey

#### 2.2.1. Gambling Expenditure (GE)

Inquiry regarding GE was approached with the question: ‘Think about the year 2016. Estimate the amount of money that you spent on gambling on average per week, per month or during the year 2016 (in euros)?’ All the responses were transformed into weekly GE.

GE was examined using two measures: weekly gambling expenditure (WGE) in euros (€) and relative gambling expenditure (RGE) (%). Weekly gambling expenditure was calculated using the formula WGE = F × GE/365.25 × 7 [20,38], where F = 30 if past-year gambling frequency was 2–3 times a month, F = 12 if past-year gambling frequency was once a month, F = 6 if past-year gambling frequency was less than monthly and F = 0 if the respondent did not gamble in 2016. For respondents who gambled on a weekly basis, GE = WGE. Overall, 34 (2.5%) of the respondents had a missing value on WGE.

RGE was estimated using WGE and the 2016 register data on disposable income, provided by Statistics Finland. RGE was calculated by dividing yearly gambling expenditure (WGE/7 × 365.25) by personal disposable income. Overall, 49 (3.6%) of the respondents had a missing value on disposable income or had value zero on disposable income.

#### 2.2.2. Gambling Participation

Gambling participation was examined by gambling frequency and number of game types. Overall gambling frequency and the number of game types were defined using a list of 18 different game types. For each game, respondents were requested to answer how often they participated in it. The frequency of gambling was classified as: less than monthly/no gambling, 1–3 times a month or at least once a week. The number of game types was classified as: 0 games, 1–2 games or 3 or more games.

### 2.3. Measures from the Register Data

The survey data were combined with the register data administered by Statistics Finland. In this study, compulsory school achievement was measured using GPA, which was retrieved from the registers. In Finland, compulsory school education ends at age 16.

For the purposes of this study, only the grades for theoretical subjects were used because of the high number of missing GPA values for grades of all subjects, and because those better reflect individuals’ reasoning, planning, abstract thinking and logical skills. In the Finnish grading system, the scores vary from 4 to 10, where 10 corresponds to outstanding, 9—very good, 8—good, 7—satisfactory, 6—mediocre and 5—passable. Based on the grading system, participants were divided into three categories: no more than mediocre (GPA 6.4 or less, corresponds to GPA 5 and 6), satisfactory to very good (GPA 6.5–9.4, corresponds to GPA 7, 8 and 9) and outstanding (GPA 9.5–10, corresponds to GPA 10).

Other measures included gender and age. Furthermore, measures of socio-economic position included highest education, employment status and disposable personal income.

#### 2.3.1. Education

Respondents were divided into two categories based on whether or not they had passed a higher education degree. Finland has two types of higher education institutions: universities and universities of applied sciences.

#### 2.3.2. Employment Status

Employment status was defined based on information from the last week of the year 2016. Respondents were divided into three categories based on whether they were employed, students or those not in education, in employment or training (NEET). The NEET category included unemployed respondents (*n* = 85), persons undergoing military or non-military service (*n* = 7), persons suffering from long-term illness (*n* = 9) and caregivers and others (*n* = 73). NEET is usually age-bound to exclude retirement-aged people and is commonly used among younger people [39].

#### 2.3.3. Disposable Personal Income

Disposable personal income obtained from register data was divided into tertiles. It was obtained by adding current transfers receivable to primary income and by deducting all current transfers payable.

### 2.4. Statistical Analysis

Respondents’ demographics, final school grade and factors related to gambling participation are presented in Table 1. A logistic regression model was conducted to examine whether past-year gambling would be more common among youth who had no more than mediocre or satisfactory to very good GPAs compared with persons with outstanding GPAs (Table 2). Persons who had not gambled during the past year were set as a reference group. In this model, past-year gambling was a dependent variable, and gender, age group, higher education degree, labour market status and disposable personal income were independent variables, which were placed into the model using the enter method. Results were presented as odds ratios (OR) and 95% confidence interval (95% CI).

To explain the variation of WGE and RGE among persons with different GPAs, two separate multiple log-linear regression models (Table 3) were conducted for past-year gamblers (*n* = 1071) because the distributions of both dependent variables were skewed to the right. Independent variables were gender, age group, higher education degree, labour market status, disposable income, GPA, gambling frequency and number of game types played. The results are presented as exponentiations of beta coefficients (exp(β)) and of 95% confidence intervals (CI). Exp(β) were interpreted as percentage differences between a subcategory and a reference category. The data were weighted based on gender, age and region of residence. All the analyses were conducted using IBM (Armonk, NY, USA) SPSS Statistics for Windows version 27.0.

The Hosmer–Lemeshow test indicated good model fit (*p* = 0.26), but Cox–Snell for logistic regression was only 0.07. However, for log-linear models, R^2^-values were much higher: 0.78 for WGE and 0.77 for RGE.

These statistical methods allow us to examine only statistical correlation between GPA and gambling participation in later life, and the results cannot be interpreted as a causal evidence of how compulsory school GPA affects gambling decisions in adulthood.

### 2.5. Ethics

The Ethics Committee of the Finnish Institute for Health and Welfare approved the research protocol (Statement THL/1390/6.02.01/2016). Potential participants were informed about the principles of voluntary participation. According to the prevailing national data protection regulations, potential participants were informed that participating in the study included links to the register. The data (without any register-based information) are available and openly accessible for research purposes from the Finnish Social Science Data Archive (FSD) with the study title of Gambling Harm Survey 2016 (ID: FSD3261; Persistent identifier: urn:nbn:fi:fsd:T-FSD3261).

## 3. Results

### 3.1. Demographics

Half (50.4%) of the respondents were women (Table 1). Nearly one in four (24.6%) had a higher education degree and over half were employed (57.3%). Most participants (81.8%) had a GPA varying from satisfactory to very good, while 7.5% had an outstanding GPA and 10.7% no more than mediocre GPA. One-fourth (25.5%) of the respondents gambled at least once a month and over half (52.1%) gambled using three or more game types.

### 3.2. Past-Year Gambling

Based on the logistic regression model, past-year gambling was more common among men than women (Table 2). Among youth who had no more than mediocre or had satisfactory to very good GPA on their primary school certificate, past-year gambling was more common compared to youths with an outstanding GPA. Furthermore, past-year gambling was more common among youth with a higher disposable income.

### 3.3. Models Explaining WGE and RGE

Based on the log-linear regression model, the male WGE was 15% higher and RGE was 13% higher than the female (Table 3). Age also had an effect on WGE; 25-year-old and older respondents spent more than 18–20-year-old respondents. However, relative to disposable income, there was no statistically significant difference between the youngest and older age groups. Those not in employment, education or training (NEET) spent 30% more money on gambling than the employed respondents. They also had higher RGE. Respondents who belonged to the highest disposable income group (18,900 € or more) spent 31% more than the lowest income group, but when examining relative gambling expenditure, they spent less than the lowest disposable income group (8900 € or less).

Both gambling participation factors had an effect on gambling expenditure. Those who gambled at least once a week spent almost eight times more than those who gambled less than once a month. This was also the case when examining relative disposable income. Engaging in three or more game types increased weekly expenditure and relative expenditure by 48 and 49%, respectively, compared to those who played 1–2 games.

Respondents without a higher education degree spent 22% more money on gambling compared to respondents with a higher education degree (Table 3). They also had higher RGE. Respondents who had no more than a mediocre GPA in the primary school certificate spent 25% more money on gambling in 2016 than those who had an outstanding GPA. When examining RGE in 2016, those with no more than a mediocre GPA spent 30% more money on gambling in 2016. When comparing respondents with an outstanding GPA, those with a satisfactory to very good GPA spent 13% more on gambling, and their RGE was 17% higher.

## 4. Discussion

Our first hypothesis was supported, as results suggest that individuals with lower compulsory school achievement (according to GPA) differ from those with a better GPA by having higher gambling expenditure later in youth. This is true even when gambling participation and socio-economic position was taken into account. Association between a low GPA and gambling expenditure persisted even though, for some respondents, many years had passed since they finished their compulsory school. It is evident that low school achievement in Finland is closely linked to later success and quality of life, such as unemployment, lower education and incomes [30]. This is mainly due to the fact that compulsory school achievement almost fully determines admission to the higher-secondary or vocational education. Thus, one explanation for the link might be that gambling in Finland is highly concentrated on individuals with a lower socio-economic position, as is suggested in our second hypothesis. This means that the low GPA can exacerbate the inequality introduced to society in the form of gambling expenditures; it has already been found by several studies that a great share of gambling expenditures come from socially disadvantaged individuals [27,29]. High gambling expenditure can be one detrimental factor of social inequality.

Further, our third hypothesis was supported, as among young people, gambling expenditure was higher among men than women, which has been shown among adults in previous studies [20,21,22].

Another possible explanation behind our finding is that the individuals with lower compulsory school achievement may possess lower cognitive abilities overall, as we have controlled many important background variables that correlate with both the GPA and gambling expenditures in our analysis, such as current education, labour market status and disposable income. The role of the cognitive abilities of an individual have recently gained attention in the literature regarding many economic outcomes, such as gambling decisions. Gong and Zhu [32] have shown that the cognitive ability of an individual is associated with gambling participation and the probability of problem gambling. They found that the higher the premorbid cognitive ability or “intelligence”, the less probable the person engages in gambling activities. Our results are qualitatively very similar to theirs: the two lower GPA groups are linked with having significantly higher gambling expenditures than the highest GPA group; in other words, higher compulsory school achievement, and possibly cognitive abilities, are related to lower gambling expenditure and, hence, lower gambling-related problems. However, the measure we use as an indicator of cognitive ability, the GPA, is not straightforwardly comparable to the test that Gong and Zhu used [32]. Consequently, one issue is that compulsory school achievement might also reflect other background factors as well as the pure cognitive ability of an individual. These (unobserved) factors include, for example, school- and family-specific factors, such as school or family resources during the compulsory school and the overall learning environment and atmosphere. Other individual factors include trauma, antisocial personality, anxiety and depression, which are all also associated with problem gambling and with poor academic achievement [40,41].

Previous evidence on preventive and harm-reduction interventions relates mostly to pre-commitment and limit setting, self-exclusion, pop-up messages and feedback [42,43]. However, youth prevention programmes have also been studied, but less than half of them have demonstrated positive effects on behaviour [42]. There has also been concern that programmes need to focus more on long-term results and the emotional aspects of gambling [44]. Many interventions focus on industry strategies with the focus on the amount of the time spent gambling and the amount of money gamblers spend [42]. Overall, previous studies are limited by lacking pre- and post-measurement, and the overall quality of the studies is weak [42,43,45]. Furthermore, differential effects of intervention strategies across socio-demographic groups have not been studied/reported. Despite this, a public health approach indicates that gambling-related harm can be reduced by intervening across the whole gambling pathway, from the regulation of access to gambling to identifying at-risk-level gamblers and services for persons with an identified gambling problem [45].

Our study contributes to the previous literature in two important ways. First, our results provide new evidence of association between the socio-economic position, cognitive abilities and gambling behaviour among youth. Our measure of compulsory school achievement (GPA), which may also reflect cognitive ability, is based on well-documented, registry-based information, which is available for a large share of Finns. In addition, as our measure of cognitive ability was measured at an earlier age, it is not affected by varying resources later in life, being more of an indicator of “natural ability”. Second, our findings also support the view that the implementation of educational or other type of preventative actions already during compulsory school, to mitigate the possible gambling-related harm in later life, might be effective regarding public health. This is an important aspect, especially in Finland, where the socially disadvantaged individuals already bear the biggest burden of gambling related harm.

We used high-quality Finnish registry data of the compulsory school GPAs and for other control variables. However, one drawback regarding our data concerns the outcome variables, which are based on a population survey with a modest participation rate. However, we have made some effort to correct this drawback by using population-weighing methods. In addition, the estimation of individuals’ own gambling expenditure and frequency can be a difficult task, which means that, with a high probability, there exists measurement error to some extent in these variables when relying on the survey-based information (see [46,47]). Further, only 7 percent of the variance of past-year gambling could be explained by the model; however, the percentages were much higher for WGE (78%) and RGE (77%). Finally, we uncovered only the statistical correlation between the GPA and the gambling decisions in later life; thus, the results cannot be interpreted as a causal evidence of how compulsory school GPA affects gambling decisions in adulthood.

## 5. Conclusions

The results supported our first hypothesis, as it was shown that lower compulsory school achievement was linked with gambling expenditure later in youth. This suggests that the gambling behaviour may be linked with the cognitive ability of an individual. Our findings also support our second and third hypothesis, and previous findings indicate that gambling is concentrated on male gender and individuals that are already socially more disadvantaged. However, it is worth noting that individual factors such as traumas, antisocial personality, anxiety and depression are all associated with gambling and with poor academic achievement. Overall, the results suggests that low compulsory school achievement can be used as way to identify young people who are at an increased risk for gambling problems later in life. Further, educational tools at a younger age can be effective in preventing gambling-related problems. Information on gambling should be given to adolescents, for example, adolescents could be taught about the logic behind the gambling games and inform them about the risks related with gambling. Special attention should be paid to boys who have difficulties with schoolwork. These results are relevant for parents, teachers and people working with adolescents.

## Figures and Tables

**Table 1 ijerph-19-09444-t001:** Demographics, final school grade and factors related to gambling participation.

Sex	%	*n*
Woman	50.4	688
Man	49.6	677
**Age group**		
18–20-year-olds	25.5	369
21–24-year-olds	30.6	505
25–29-year-olds	43.9	460
**Final school grade, GPA ^2^**		
Outstanding (9.45–10)	7.5	74
Satisfactory to good (6.45–9.44)	81.8	960
No more than mediocre (6.44 or less)	10.7	139
**Higher education degree**		
Yes	24.6	321
No	75.4	1013
**Labour market status ^1^**		
Employed	57.3	750
Student	29.9	426
NEET	12.8	158
**Disposable income ^1^**		
≤8900 €	33.2	450
9000–18,800 €	33.3	449
≥18,900 €	33.5	388
**Gambling frequency ^1^**		
At least once a week	18.0	212
1–3 times a month	25.5	328
Less often	35.8	490
Not in 2016	20.7	294
**Number of game types gambled ^1^**		
0 games	21.4	306
1–2 games	26.5	365
3 or more games	52.1	663

The weighted data (%), *n* = 1334 (non-weighted); Percentages are calculated from the weighted data, frequencies from the non-weighted data; NEET, not in employment, education or training; ^1^ in 2016; ^2^ GPA, grade point average incl. the grades for theoretical subjects, compulsory school.

**Table 2 ijerph-19-09444-t002:** Logistic regression model explaining past-year gambling. Results presented as odds ratios (OR) and 95% confidence interval (95% CI).

	Past-Year Gambling ^1^	
	OR	95% CI
**Sex**		
Woman	1	1
Man	1.77 ***	1.29–2.45
**Age group**		
18–20-year-olds	1	1
21–24-year-olds	0.85	0.55–1.33
25–29-year-olds	0.83	0.50–1.37
**Final school grade, GPA ^2^**		
Outstanding (9.45–10)	1	1
Satisfactory to very good (6.45–9.44)	2.98 *	1.18–7.52
No more than mediocre (6.44 or less)	1.63 *	1.06–2.51
**Higher education degree**		
Yes	1	1
No	1.48	0.97–2.25
**Labour market status ^3^**		
Employed	1	1
Student	0.71	0.47–1.06
NEET	1.26	0.74–2.16
**Disposable income ^4^**		
≤8900 €	1	1
9000–18,800 €	1.89 **	1.27–2.82
≥18,900 €	4.07 ***	2.34–7.06

The weighted data, *n* = 1334 (non-weighted); ^1^ in 2016; ^2^ GPA, grade point average incl. the grades for theoretical subjects, compulsory school; ^3^ NEET, not in education, employment or training; ^4^ tertiles of personal disposable income based on register data; Reference category: no past-year gambling; * ≤0.05, ** ≤0.01, *** ≤0.001.

**Table 3 ijerph-19-09444-t003:** Log-linear models explaining weekly (WGE) and relative gambling expenditure (RGE).

	Weekly Gambling Expenditure ^1^ (€)		Relative Gambling Expenditure ^1^ (%)	
	Exp(β)	95% CI	Exp(β)	95% CI
**Sex**				
Woman	1	1	1	1
Man	1.15 ***	1.07–1.23	1.13 ***	1.05–1.23
**Age group**				
18–20-year-olds	1	1	1	1
21–24-year-olds	1.09	0.98–1.22	0.97	0.86–1.09
25–29-year-olds	1.16 **	1.03–1.31	1.01	0.89–1.15
**Final school grade, GPA ^2^**				
Outstanding (9.45–10)	1	1	1	1
Satisfactory to good (6.45–9.44)	1.13 *	0.98–1.28	1.17 *	1.03–1.34
No more than mediocre (6.44 or less)	1.25 **	1.05–1.49	1.30 **	1.07–1.57
**Higher education degree**				
Yes	1	1	1	1
No	1.22 ***	1.11–1.33	1.23 ***	1.12–1.36
**Labour market status ^1^**				
Employed	1	1	1	1
Student	1.08	0.98–1.21	1.24 ***	1.11–1.38
NEET	1.30 ***	1.16–1.47	1.38 ***	1.22–1.57
**Disposable income ^1^**				
≤8900 €	1	1	1	1
9000–18,800 €	1.08	0.98–1.20	0.69 ***	0.62–0.77
≥18,900 €	1.31 ***	1.16–1.49	0.67 ***	0.59–0.77
**Gambling frequency ^1^**				
Less than once a month	1	1	1	1
1–3 times a month	2.86 ***	2.61–3.10	2.87 ***	2.62–3.15
At least once a week	7.85 ***	6.96–8.50	7.81 ***	7.01–8.69
**Number of game types gambled ^1^**				
1–2 games	1	1	1	1
3 or more games	1.48 ***	1.36–1.61	1.49 ***	1.36–1.62

The weighted data, *n* = 1071 (non-weighted); ^1^ in 2016; NEET, not in employment, education or training; ^2^ GPA, grade point average incl. the grades for theoretical subjects, compulsory school. * ≤0.05, ** ≤0.02, *** ≤0.001.

## Data Availability

The survey data, without any register-based information, are publicly accessible for research purposes from the Finnish Society Science Data Archive (FSD), with the name of Gambling Harm Survey 2016 (ID: FSD3261; Persistent identifier: urn:nbn:fi:fsd:T-FSD3261).

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
