# Peer review of "Compulsory School Achievement and Future Gambling Expenditure: A Finnish Population-Based Study"

_ijerph, 2022, doi:10.3390/ijerph19159444_

Round 1

Reviewer 1 Report

Dear Authors,

Thank you for the extensive review and refinement of the language, analysis and conclusions. 

I have two small recommendations. Please add citations to your last sentence, paragraph 1, page 9, discussing individual factors associated with problem gambling.

Page 2, paragraph 3, line 71: The text "(Deary, 2006, Intelligence and educational achievement)." needs to be deleted.

Congratulations on your publication.

Author Response

We would like to thank once again all the reviewers for their valuable comments. We have tried to address all the concerns you have raised to our best ability.

Reviewer 1:

I have two small recommendations. Please add citations to your last sentence, paragraph 1, page 9, discussing individual factors associated with problem gambling.

Response: This is very good point, and two citations are added: Kee-Lee Chou, Tracie O. Afifi, Disordered (Pathologic or Problem) Gambling and Axis I Psychiatric Disorders: Results From the National Epidemiologic Survey on Alcohol and Related Conditions, American Journal of Epidemiology, Volume 173, Issue 11, 1 June 2011, Pages 1289–1297, https://doi.org/10.1093/aje/kwr017

Bruffaerts R, Mortier P, Kiekens G, Auerbach RP, Cuijpers P, Demyttenaere K, Green JG, Nock MK, Kessler RC. Mental health problems in college freshmen: Prevalence and academic functioning. J Affect Disord. 2018 Jan 1;225:97-103. doi: 10.1016/j.jad.2017.07.044.

Page 2, paragraph 3, line 71: The text "(Deary, 2006, Intelligence and educational achievement)." needs to be deleted.

Response: Thank you for noticing this! This is deleted as suggested.

Congratulations on your publication.

Response: Thank you!

Reviewer 2 Report

Thank you for the opportunity to review your article. It looks as though I am reviewing a copy that has been edited. I believe the parts that were edited were important aspects needing addressing in perhaps the first article draft submitted.

I would like to see more consistency from beginning to end in all the points you discuss in your introduction. You discussed differing behaviors and GE of men and women in your introduction. This was then not part of one of your hypotheses for this study. You reviewed the results of men versus women in your Results section. Then again, it was not wrapped up in the Conclusion and Discussion. I was confused by why this was there and then not consistently discussed in all aspects of your article. I believe you could add as hypothesis one that your study would confirm previous findings that men and women would differ.

I did not see evidence that you controlled for gender when specifically testing your stated hypotheses. I would be interested in seeing statistical analysis of these two original hypotheses with gender controlled.

Author Response

We would like to thank once again all the reviewers for their valuable comments. We have tried to address all the concerns you have raised to our best ability.

Reviewer 2:

Thank you for the opportunity to review your article. It looks as though I am reviewing a copy that has been edited. I believe the parts that were edited were important aspects needing addressing in perhaps the first article draft submitted.

I would like to see more consistency from beginning to end in all the points you discuss in your introduction. You discussed differing behaviors and GE of men and women in your introduction. This was then not part of one of your hypotheses for this study. You reviewed the results of men versus women in your Results section. Then again, it was not wrapped up in the Conclusion and Discussion. I was confused by why this was there and then not consistently discussed in all aspects of your article. I believe you could add as hypothesis one that your study would confirm previous findings that men and women would differ.

Response: Thank you for this good point! Hypothesis 3 is added, and more discussion is added to discussion at page 9 and conclusion sections at page 11 on gender differences.

I did not see evidence that you controlled for gender when specifically testing your stated hypotheses. I would be interested in seeing statistical analysis of these two original hypotheses with gender controlled.

Response: In these models gender is adjusted, but because of the small number of women gamblers we are not able to study males and females separately.

Reviewer 3 Report

I appreciate the subject matter undertaken by the Authors. I would like to stress at the beginning that the article under review presents high scholar standards. The subject at stake is interesting both from the scientific and practical point of view, as well. Gambling plays a very relevant role in terms of legal, social, political and public health reasons. It is a source of many risks and threats at many different levels. This is particularly crucial amid youth addiction issues. All data collected and presented are vital and important. I agree with all theses and conclusions put forward by the Authors. However, I suggest that Authors should make reference to the latest article by José Miguel Gimenez Lozano and Francisco Manuel Morales Rodríguez, Systematic Review, Preventive Intervention to Curb the Youth Online Gambling Problem, Sustainability 2022, 6402; https://doi.org/10.3390/su14116402. In this publication under review, there is a statement that “For example, adolescents could be taught about logic behind the gambling games and inform them about the risks related with gambling” (p. 10, marginal ref. number: 341-342). So, it might mean, that certain preventive measures regarding youths can be realized by even teachers or other educators (question whether parents can be involved, too ?), meaning persons without any special preparation. Gimenez-Lozan and  Rodríguez come to different conclusion. They claim that such measures may be carried by certain specialists – professionals, like one may say psychologists and therapists. Anyway, those shall be specially educated and trained specialists. In my opinion, the problem of youth and their addiction to gambling is so important, that it requires very careful and thoughtful approach. The Authors of the article under review should consider those remarks. On the other hand, I appreciate a lot their attempt to find a connection between personal characteristics of given persons and their propensity to gambling.

Additionally, I suggest changing the title of this article, as to make it shorter and more essential. Now, there is an evident thesis in it presented. I may suggest following title: "Compulsory school achievement and future gambling expenditure. A Finnish population-based study".

Author Response

We would like to thank once again all the reviewers for their valuable comments. We have tried to address all the concerns you have raised to our best ability.

Reviewer 3:

I appreciate the subject matter undertaken by the Authors. I would like to stress at the beginning that the article under review presents high scholar standards. The subject at stake is interesting both from the scientific and practical point of view, as well. Gambling plays a very relevant role in terms of legal, social, political and public health reasons. It is a source of many risks and threats at many different levels. This is particularly crucial amid youth addiction issues. All data collected and presented are vital and important. I agree with all theses and conclusions put forward by the Authors. However, I suggest that Authors should make reference to the latest article by José Miguel Gimenez Lozano and Francisco Manuel Morales Rodríguez, Systematic Review, Preventive Intervention to Curb the Youth Online Gambling ProblemSustainability 2022, 6402; https://doi.org/10.3390/su14116402. In this publication under review, there is a statement that “For example, adolescents could be taught about logic behind the gambling games and inform them about the risks related with gambling” (p. 10, marginal ref. number: 341-342). So, it might mean, that certain preventive measures regarding youths can be realized by even teachers or other educators (question whether parents can be involved, too ?), meaning persons without any special preparation. Gimenez-Lozan and  Rodríguez come to different conclusion. They claim that such measures may be carried by certain specialists – professionals, like one may say psychologists and therapists. Anyway, those shall be specially educated and trained specialists. In my opinion, the problem of youth and their addiction to gambling is so important, that it requires very careful and thoughtful approach. The Authors of the article under review should consider those remarks. On the other hand, I appreciate a lot their attempt to find a connection between personal characteristics of given persons and their propensity to gambling.

Response: Thank you for that interesting reference! We added it to references.

We agree that in prevention programs which specially aims to prevent gambling addiction, specialists are needed. In our sentence “For example, adolescents could be taught about logic behind the gambling games and inform them about the risks related with gambling” the aim was, that giving basic information on gambling and possible threads associated with it, would be beneficial to adolescents. This is re-worded as following: Information on gambling should be given to adolescents, for example, adolescents could be taught about logic behind the gambling games and inform them about the risks related with gambling.

Additionally, I suggest changing the title of this article, as to make it shorter and more essential. Now, there is an evident thesis in it presented. I may suggest following title: "Compulsory school achievement and future gambling expenditure. A Finnish population-based study".

Response: That is good idea! Title is changed as suggested.

This manuscript is a resubmission of an earlier submission. The following is a list of the peer review reports and author responses from that submission.

Round 1

Reviewer 1 Report

This paper provides a description to the link between school achievement and gambling behaviour. Specifically, it analyse the association of GPA and a number of variables on gambling including gambling participation, and gambling expenditure of the gamblers. Simple regressions are conducted.

Comments/questions:

  1. The regressions specified in this paper are descriptive and do not reflect causal relationships between the dependent and the explanatory variables. This has to be made clear as it affects the conclusions.
  2. Presentation could be improved:
    • Tables 2 and 3 are not so clearly presented: The use of exp(beta) is odd. For the logistic regressions, either the parameter themselves, marginal effects at the mean, or something  similar could be presented. For the linear regression, the parameters should be presented. It is unclear what the 95% CIs are for, beta's or exp(beta)'s? ; It is not clear which levels of significance the symbols of *  correspond to.
    • Data description can be clearer: e.g., Second paragraph in Page 3 is confusing. For example, it is unclear 71% is out of what? while it is said that the sample size is 19933 persons, it is also said that 7186 adults participated in the study. It is stated here that for this study, the sample used for analysis  is 1365, but in Table 1, only 1334 individuals are used. The difference is not spelt out.
    • Full names should be given when acronyms are first used.
    • No explanations are given for the variables created or used in the analysis, for example why GPA of 6.44 or less is regarded as 'mediocre', or where this criteria is obtained, while 7 was defined as 'satisfactory'. Similarly, how high, middle, or low income are defined?
    • Style of reference are inconsistent. Sometimes they are numbered, and some other times author/year are used. The reference list is not in alphabetic order.
  3. Perhaps, attention needs to be paid to model specifications and variable definitions. For example, employment status, is defined as employed, students, and other not-employed (the writing in Page 4 is not clear enough). Why are individuals partitioned into these three categories? What is the purpose? Gamble frequency and no. of game types gambled are included in the expenditure equations. But at least GE was calculated using information of gamble frequency. Can you explain something using the same information? What the results really tell you?

Reviewer 2 Report

Dear Authors,

You have presented an interesting paper that can make a valuable contribution to the literature. It is overall a good paper that can be improved with some revision.

A minor point is that there are some English language errors, including typographical and grammatical errors. These errors do not generally confuse or mislead the reader, however a careful review is worthwhile. For example, I believe that 'registrars' on page 3, line 99 should be 'registers'. 

I have two more substantial concerns that I feel should be revised before the paper is ready for publication.

The first is a request for clarification of the raw data collected to calculate disposable income variable, and a potential limitation of this data. I believe this is discussed on page 3, line 124, and I have the impression that expendable income was not an estimate of each participants' disposable income; instead it was an estimate of the disposable income of this age group of the population? Is that correct? If that is the case, then that is a clear limitation of the variable, which must be discussed in the limitations section of the Discussion. A more detailed explanation of this variable would be greatly appreciated.

The second is the Authors' conclusion that the association between poor academic achievement and gambling expenditure may be a reflection of 'lower cognitive ability'. While this is a possible explanation, it is also worth noting that childhood neglect, other traumas, grief, antisocial personality, anxiety and depression are all also associated with problem gambling and with poor academic achievement. This is an essential discussion to be made and acknowledged in the conclusion and abstract. Furthermore, it can inform a more considered suggestion for gambling harm prevention.

A suggestion for your consideration is the potential to conduct an analysis that specifically controls for certain variables (e.g., demographics, gambling product) to directly answer the question of whether GPA can predict expenditure after controlling for these factors.

I look forward to your reply and revisions.

Kind regards.

Reviewer 3 Report

The topic of the article is important and interesting. However:

  • there are no hypotheses
  • the literature review is rather poor, as discussion, where authors even discuss with themselves, rather than more broadly, with other authors
  • in line 83 "as grade point averages (GPA)" authors explain the acronym GPA, however, use this acronym earlier (line 74)
  • it is hard to follow what was the size of the research sample. Also, the way of presenting it is hard to follow. For example, line 104 "Participants chose more often online survey (71%, n = 5,084)", whereas in line 106 "leading the study sample size to 19,933 persons" and then in line 109 "only 18-29-year-old respondents were selected (N=1,365)." Thus it should be everything should be presented in a different order, firstly what was the general sample size of the research, and then for the present study,
  • the models are not well described. What were the dependent variables? It should be described in the methodology not somewhere below the tables. Is for example regression model well fitted to the data? What was the value of the Hosmer-Lemeshow test, Cox-Snell? What was the method of eliminating the potential independent variables to model? Wald elimination or something else? Why the data were weighted? 
  • Line 213 " 18–21-year-old respondents." Should be 18-20.
  • Line 216 and 217 ". Respondents who belonged to the highest disposable income group 216
    (19,300 € or more) spent 31% " - in the table highest  disposable income is "≥ 18,900€"
  • Line 213 and 214 "However, relative to disposable income, 213
    there were no differences between the age groups. " - but the authors didn't interactions between variables that could allow drawing such conclusions;
  • The discussion with other studies is rather poor,
  • In the conclusion the authors don't refer to hypotheses (because no hypotheses); the limitations of the study should be here not in discussion; there is no information on who could benefit from the survey results, for example, which institutions.